# Functionalization of Multi-Walled Carbon Nanotubes Changes Their Antibiofilm and Probiofilm Effects on Environmental Bacteria

**DOI:** 10.3390/microorganisms10081627

**Published:** 2022-08-11

**Authors:** Yuliya Maksimova, Yana Bykova, Aleksandr Maksimov

**Affiliations:** 1Laboratory of Molecular Biotechnology, Institute of Ecology and Genetics of Microorganisms UB RAS, Perm 614081, Russia; 2Department of Microbiology and Immunology, Perm State University, Perm 614990, Russia

**Keywords:** multi-walled carbon nanotubes, microbial biofilms, activated sludge, actinobacteria, proteobacteria

## Abstract

Releasing multi-walled carbon nanotubes (MWCNTs) into ecosystems affects the biofilm formation and metabolic activity of bacteria in aquatic and soil environments. Pristine (pMWCNTs), oleophilic (oMWCNTs), hydrophilic (hMWCNTs), and carboxylated (cMWCNTs) carbon nanotubes were used to investigate their effects on bacterial biofilm. A pronounced probiofilm effect of modified MWCNTs was observed on the Gram-negative bacteria of *Pseudomonas fluorescens* C2, *Acinetobacter guillouiae* 11 h, and *Alcaligenes faecalis* 2. None of the studied nanomaterials resulted in the complete inhibition of biofilm formation. The complete eradication of biofilms exposed to MWCNTs was not observed. The functionalization of carbon nanotubes was shown to change their probiofilm and antibiofilm effects. Gram-negative bacteria were the most susceptible to destruction, and among the modified MWCNTs, oMWCNTs had the greatest effect on biofilm destruction. The number of living cells in the biofilms was assessed by the reduction of XTT, and metabolic activity was assessed by the reduction of resazurin to fluorescent resorufin. The biofilms formed in the presence of MWCNTs reduced tetrozolium to formazan more actively than the control biofilms. When mature biofilms were exposed to MWCNTs, dehydrogenase activity decreased in *Rhodococcus erythropolis* 4-1, *A. guillouiae* 11 h, and *A. faecalis* 2 in the presence of pMWCNTs and hMWCNTs, as well as in *A. guillouiae* 11 h exposed to cMWCNTs. When mature biofilms were exposed to pMWCNTs, hMWCNTs, and cMWCNTs, the metabolism of cells decreased in most strains, and oMWCNTs did not have a pronounced inhibitory effect. The antibiofilm and probiofilm effects of MWCNTs were strain-dependent.

## 1. Introduction

Carbon nanotubes (CNTs) are a relatively new material with unique properties that are now widely used in various industries [1,2]. Large amounts of production and intensive use lead to the release of these nanomaterials into the environment, wastewater, and soil. The effect of carbon nanotubes on bacteria has been widely studied, both in connection with their cytotoxic effect [3,4] and in connection with the use of nanotubes in biotechnology, in particular, in microbial fuel cells [5].

The antimicrobial properties of nanotubes depend on many factors, including size, functionalization, and dispersion. Smaller particles have a higher surface-to-volume ratio and, therefore, can easily attach to microbial cells and affect membranes and metabolic processes [6]. In this regard, multi-walled carbon nanotubes (MWCNTs) seem to be less toxic to bacteria compared to single-walled carbon nanotubes (SWCNTs). MWCNTs have a larger diameter, higher inherent rigidity, and possibly smaller van der Waal’s forces at the surface. All of this reduces the toxicity of MWCNTs towards bacterial cells. At the same time, thin MWCNTs with a smaller diameter and length have a stronger antibacterial effect [7].

The functionalization of MWCNTs can significantly change the toxic properties of nanotubes with respect to bacteria. The properties of CNTs are changed by carboxylation, amidation, hydroxylation, and conjugation with antibiotics and metals, including metal nanoparticles. MWCNT conjugated with copper nanoparticles increased the antimicrobial activity against *Methylobacterium* spp. biofilms by cell wall degradation via direct the contact and inhibition of quorum sensing [8]. Functionalization with arginine and lysine enhanced the antibacterial activity of MWCNTs, especially against Gram-negative bacteria [9]. Modification with functional groups, metals, and various organic substances can directly affect the bacterial cell. In addition, the modification of the surface of MWCNTs significantly affects the dispersion of nanotubes, and the damaging effect on cells, in turn, depends on the dispersion [10].

The introduction of various CNTs into the environment can significantly change the structure of the microbial community. It was noted that in a microbial community of activated sludge, the rate of oxygen uptake, the rate of nitrification and denitrification, and dehydrogenase activity were decreased with increasing concentration of MWCNTs-OH. The change in the Shannon and Simpson indexes showed that the microbial diversity decreased [11]. At the same time, it was shown that MWCNTs not only did not destroy oil-degrading bacteria but also led to an increase in their growth in oil-polluted areas [12]. The impact of MWCNTs on the microbial community in sediments contaminated with 2,4-dichlorophenol was highly dependent on the concentration of nanoparticles. The microbial biomass increased in the presence of 5 mg/g of MWCNTs, but did not significantly change at other MWCNT concentrations [13].

Bacteria are known to exist in nature mainly in the form of biofilms, i.e., multispecies communities immersed in an extracellular polymeric matrix (EPS) [14,15]. Cells in biofilms are much more resistant to various adverse physicochemical effects, while the level of adaptive response may differ from planktonic cells. There is evidence that cells in the biofilm structure are less sensitive to the effects of CNTs, and the EPS plays a key role in reducing the adverse effects of CNTs [16]. CNTs can be incorporated into EPS, which prevents the direct contact of nanoparticles with cell membranes [17].

Natural bacterial biofilms can have various implications for humans. These can be corrosive biofilms that cause the destruction of materials, pipelines, and structures for various purposes and biofilms on medical equipment and catheters, which are a source of infection that is difficult to treat. On the other hand, these can be biofilms with functions useful for humans, for example, self-immobilized bacterial communities of activated sludge [18,19,20], biofilms of electrochemically active microorganisms in microbial fuel cells [21,22,23], and biocatalytic biofilms [24,25,26]. Microbial biofilms are useful in environmental treatment technologies [27]. These biofilms have the ability to biodegrade toxic substances, polyaromatic hydrocarbons [28], petroleum products [29], and pesticides [30].

Biofouling control can have several directions, one of which consists of the development of various composite materials. Microbial adhesion and biofilm formation are strongly influenced by surface topography [31], and the surface of composite materials can be modified by incorporating CNTs into their composition. For example, a nanoporous solid-state aluminum membrane modified with carboxylated MWCNTs exhibited antimicrobial and antiadhesive properties against *Escherichia coli* and *Staphylococcus aureus*; the number of killed bacteria on such membranes exceeded 98%. Despite the fact that cells were adsorbed on the surface of the nanotubes, contact with carboxylated MWCNTs led to cell destruction, and the membranes self-cleaned when the dead cells were washed away with a stream of water [32]. Antimicrobial nanocomposite MWNT-epilsonpolylysine had improved antimicrobial activities and antiadhesive activity against *E. coli, Pseudomonas aeruginosa*, and *S. aureus* [33]. It was shown that a low-density polyethylene nanocomposite containing 1% MWCNTs reduced the adhesion of *E. coli* and showed a biocidal effect on it [34]. The development of biomaterials capable of preventing biofilm-related infections is necessary for several medical applications, including the construction of medical devices and implants [33,35]. Vertically aligned immobilized MWCNTs have been shown to manifest strong anti-adhesive effects on the biofilms of human pathogens *Klebsiella pneumoniae, K. oxytoca, P. aeruginosa*, and *S. epidermidis* [36]. Biofouling in the aquatic environment is also a big problem, for which it is necessary to develop new materials with antibiofilm activity. To reduce biofouling, composite materials are designed with CNTs in their composition. MWCNTs in the chlorinated rubber matrix are known to significantly improve the antibiofouling effects of this material in the marine environment and reduce the attachment and colonization of the surface of pioneer eukaryotic microbes [37].

Thus, the introduction of MWCNTs into composite materials can increase their antimicrobial and antifouling activity. However, MWCNTs can have different effects on biofilms depending on the form in which the nanomaterial is present in the medium. Less attention has been paid to studying the effect of dispersed CNTs on biofilms. At the same time, when released into wastewater, carbon nanotubes will be in a dispersed or partially aggregated state, which makes it necessary to study the effect of such CNTs on the biofilms of the bacterial community, primarily the activated sludge community. At present, there are sufficient data on the effect of MWCNTs on bacterial cells in suspension. Biofilm is a natural state of environmental bacteria. The response of cells to external influences and their adaptive responses in suspension and biofilm are different, and therefore, it is necessary to study the effect of nanomaterials on microbial biofilms.

Actinobacteria and proteobacteria are important representatives of environmental microbiomes, soils, and waters; they play a significant role in the processes of bioremediation, degradation of pollutants, and wastewater treatment [38,39]. The ubiquitous bacteria of the genus *Rhodococcus* are typical representatives of soils and have a high biodegradative potential. The bacteria of *Pseudomonas, Achromobacter, Alcaligenes, Acinetobacter,* and *Burkholderia* genera possess enzymatic systems for the degradation of organic pollutants and have been found in activated sludge, river silts, and water. Therefore, in this investigation, these objects were chosen to study the effect of pristine and modified MWCNTs on biofilm formation and the metabolic activity of bacterial cells in biofilms. The aim of this investigation was to study the effect of functionalized MWCNTs on biofilm formation and the viability of proteobacteria of the genera *Achromobacter, Acinetobacter, Alcaligenes, Burkholderia, Pseudomonas*, and actinobacteria *Rhodococcus* in biofilms.

## 2. Materials and Methods

### 2.1. Strains and Culture Medium

*Achromobacter pulmonis* PNOS and *Burkholderia dolosa* BOS were previously isolated on pyridine as the only carbon source [40] from the activated sludge of treatment facilities in Perm (Russia). These strains are active degraders of pyridine. *Alcaligenes faecalis* 2 and *Acinetobacter guillouiae* 11 h were isolated on 3-cyanopyridine from the activated sludge of the same treatment facilities [41]. These strains have amidase activity. *Pseudomonas fluorescens* C2 (VKM V-2597D) with nitrilase activity was isolated from soil contaminated with acrylonitrile [42], and *Rhodococcus erythropolis* 11-2, *R. erythropolis* 4-1, *R. erythropolis* IL BIO, *R. ruber* gt1 (IEGM 612, Regional profiled collection of alkanotrophic microorganisms, acronym IEGM, www.iegm.ru/iegmcol/index.html, accessed on 8 June 2022) [43] were isolated from the soils of Perm Kray (Russia). These strains exhibit nitrile hydratase activity.

### 2.2. Multi-Walled Carbon Nanotubes

The MWCNTs “Taunit-M” (“NanoTechCenter” Ltd., Tambov, Russia) had the following characteristics: external diameter of 10–30 nm, internal diameter of 5–15 nm, length ≥ 2 µm, specific surface area ≥270 m^2^ g^−1^, and bulk density of 0.025–0.06 g cm^−3^. Pristine MWCNTs (pMWCNTs) and functionalized MWCNTs were used. The functionalized MWCNTs had the following characteristics: external diameter of 20–50 nm, internal diameter of 10–20 nm, length ≥ 2 µm, and specific surface area ≥160 m^2^ g^−1^. The oleophilic MWCNTs (oMWCNTs) were modified by fatty acid residues up to 15 wt. %. The hydrophilic MWCNTs (hMWCNTs) contained hydroxyl and carboxyl groups as a result of the oxidation of the carbon nanotubes with sodium hypochlorite in an alkaline aqueous solution and had water solubility up to 0.2%. The carboxylated MWCNTs (cMWCNTs) had 0.1–1.0 mmol/g COOH groups. 

### 2.3. Determination of Biofilm Biomass

Biofilms were grown for 7 days in the wells of a 96-well plate in 200 µL of Luria-Bertani (LB) medium inoculated with 5 µL of a bacterial suspension containing (1.5–1.7 × 10^9^) CFU/mL. The LB medium contained 200 mg/L MWCNTs, and the LB medium without MWCNTs served as a control. To obtain a homogeneous suspension, the medium with MWCNTs was preliminarily treated with ultrasound in the Elma Ultrasonic 30S bath, Elma (Germany) at 37 kHz 10 times for 1 min. Planktonic cells were removed from the wells by decantation, the biofilm was washed twice with 200 µL of potassium phosphate buffer, and the biofilm biomass was determined. The biofilm was stained with 0.1% crystal violet for 40 min in the dark. Then, the staining agent was removed, the stained biofilm was washed once with potassium phosphate buffer, and the staining agent was extracted with 96% ethanol (200 µL). Biofilm formation was assessed by the optical density of the staining solution at 540 nm using an Infinite M1000 Pro plate reader (Tecan, Männedorf, Switzerland).

### 2.4. Biofilm Eradication in the Presence of pMWCNTs and Functionalized MWCNTs

Biofilms were grown for 7 days in the wells of a 96-well plate in 200 μL of LB medium inoculated with 5 μL of a bacterial suspension containing (1.5–1.7 × 10^9^) CFU/mL. Planktonic cells were removed from the wells by decantation, the biofilm was washed twice with 200 μL of potassium phosphate buffer, and 0.9% NaCl solution with 200 mg/L MWCNTs or 0.9% NaCl solution (control) was added. After incubation at 30 °C for 24 h, the supernatant was removed and the biofilms were stained as described previously.

### 2.5. Assessment of Respiratory Activity of Biofilms

*A. faecalis* 2, *A. guillouiae* 11 h, and *P. fluorescens* C2 biofilms were grown within 3 days, and *R. erythropolis* 11-2, *R. erythropolis* ILBIO, *R. erythropolis* 4-1, and *R. ruber* gt1 biofilms were grown within 7 days in the wells of a 96-well plate in 200 µL of LB medium inoculated with 5 µL of a bacterial suspension (1.5–1.7 × 10^9^) CFU/mL. Planktonic cells were removed from the wells by decantation, the biofilm was washed twice with 200 µL of potassium phosphate buffer, and 100 µL of 0.9% NaCl containing 200 mg/L MWCNTs or 0.9% NaCl was added as a control. After 1 h of exposure, the supernatant was removed and 100 µL of 0.9% NaCl with 50 µL of the XTT reagent (ApplChem GmbH, Darmstadt, Germany) was added. Sodium 3,3′-[1-[(phenylamino)carbonyl]-3,4-tetrazolium]Bis(4-methoxy)-6-nitro)benzene sulfonic acid hydrate (XTT), during aerobic bacterial metabolism, was reduced to a soluble colored formazan compound [44]. The optical density at 480 nm was measured for 7 h using an Infinite M1000 Pro plate reader (Tecan, Männedorf, Switzerland). In another case, measurements were performed in the presence of CNTs, while 100 µL of 0.9% NaCl containing 200 mg/L MWCNTs and 50 µL of the XTT reagent were added to the biofilms simultaneously.

In the second variant of experiments, biofilms of *A. faecalis* 2, *A. guillouiae* 11 h, and *P. fluorescens* C2 were grown on LB medium containing 200 mg/L MWCNTs for 3 days, and biofilms of *R. erythropolis* 11-2, *R. erythropolis* ILBIO, *R. erythropolis* 4-1, and *R. ruber* gt1 for 7 days on LB medium without MWCNTs as a control. The biofilms were washed twice with 200 μL of potassium phosphate buffer and 100 μL of 0.9% NaCl solution and 50 μL of XTT reagent were added. The optical density at 480 nm was determined as described above.

The coefficient of the MWCNT effect on respiration (I_deg_) was determined as the ratio of the optical density OD_exp_ obtained during the interaction of the cells with the XTT reagent after exposure to the MWCNTs to the OD_cont_ obtained for the control samples. The effect of MWCNTs on respiratory activity was assessed using I_deg_. The MWCNTs led to an increase in the intensity of respiration at I_deg_ > 1, they led to a decrease in the intensity of respiration at I_deg_ < 1, and there was no effect at I_deg_ = 1. The OD was measured over 6 h in one-hour intervals.

### 2.6. Determination of Metabolic Activity of Biofilm Cells

The metabolic activity of biofilm cells was assessed by fluorescence after staining with PrestoBlueTM Cell Viability Reagent (Thermo Fisher Scientific, Waltham, MA, USA). For this, cultures were grown in 100 µL of LB medium in a 96-well black plate (Nunc, Roskilde, Denmark) as described above, plankton cells were removed, and the biofilms were washed once with 0.9% NaCl solution. Then, 100 µL of 0.9% NaCl solution with MWCNTs was added, and it was incubated for 1 h at 25 °C. The supernatant was removed and 90 µL of 0.9% NaCl solution and 10 µL of staining agent were added to each well, and it was incubated for 10 min at 37 °C. Then, the fluorescence was measured at λ 560/590 (Ex/Em) by an Infinite M1000 Pro plate reader (Tecan, Männedorf, Switzerland).

### 2.7. Statistical Analysis

The results obtained were processed statistically, and the means, standard deviations, and confidence intervals were determined. The significance of differences was assessed using Student’s *t*-test, *p* < 0.05 (n = 7–14).

## 3. Results

### 3.1. Effects of pMWCNTs and Functionalized MWCNTs on Bacterial Biofilm Formation

This section is devoted to the investigation of the effect of pristine and functionalized MWCNTs on bacterial biofilm formation (Figure 1).

It was shown that, compared to the control, pMWCNTs caused a significant increase in the biofilm formation of *R. erythropolis* ILBIO, *P. fluorescens* C2, *A. guillouiae* 11 h, and *A. faecalis* 2. oMWCNTs caused an increase in the biofilm formation of *R. erythropolis* ILBIO, *P. fluorescens* C2, *A. guillouiae* 11 h, *A. faecalis* 2, and *B. dolosa* BOS. hMWCNTs caused an increase in the biofilm formation of *P. fluorescens* C2, *A. guillouiae* 11 h, *A. faecalis* 2, and *B. dolosa* BOS. cMWCNTs caused an increase in the biofilm formation of *R. ruber* gt1, *P. fluorescens* C2, *A. guillouiae* 11 h, *A. faecalis* 2, and *B. dolosa* BOS.

Compared to pMWCNTs, the biofilm formations of *A. guillouiae* 11 h and *A. faecalis* 2 significantly decreased under the effect of all the studied functionalized MWCNTs, and *A. pulmonis* decreased under the effect of oMWCNTs. The biofilm formation of *P. fluorescens* C2 significantly increased under the influence of all studied nanotubes. The biofilm biomass of *A. pulmonis* PNOS increased under the influence of cMWCNTs compared to pMWCNTs, and *B. dolosa* BOS increased under the influence of cMWCNTs and hMWCNTs. In rhodococci, there were no regularities in the biofilm formation with modified MWCNTs in the medium; cMWCNTs increased the biofilm formation of *R. ruber* gt1 and *R. erythropolis* 11-2, but decreased the biofilm biomass of *R. erythropolis* ILBIO. Gram-negative bacteria were most susceptible to the action of pristine and functionalized MWCNTs. *A. guillouiae* 11 h and *A. faecalis* 2, in the presence of pristine MWCNTs, formed larger biofilms compared to those in LB medium. In the media with modified MWCNTs, the biofilm biomasses decreased compared to those with pMWCNTs. In *P. fluorescens* C2, on the contrary, the biofilm biomass in the presence of pMWCNTs was lower than that in the control, but with the functionalization of MWCNTs, it was significantly higher than in the control and in the presence of pMWCNTs. Compared to the control, in *R. erythropolis* ILBIO, the biofilm biomass significantly increased only under the effect of pMWCNTs and oMWCNTs, and in *R. ruber* gt1, the biomass increased under cMWCNTs. The biomass of the *R. erythropolis* 11-2 biofilm slightly decreased under the effect of pMWCNTs and oMWCNTs, and the biofilm biomass of *R. erythropolis* 4-1-decreased under cMWCNTs. Of all the strains of Gram-negative bacteria, only *P. fluorescens* C2 under the effect of pMWCNTs and *A. pulmonis* PNOS under the effect of oMWCNTs showed a slight decrease in biofilm biomass; in all other cases, the studied Gram-negative bacteria formed greater biofilm biomass. The biofilm formation of Gram-negative bacteria, as compared to Gram-positive bacteria, was more affected by MWCNTs in the medium; the biofilm biomass of Gram-negative bacteria was 4 times greater than that of the control.

### 3.2. Effects of pMWCNTs and Functionalized MWCNTs on Biofilm Eradication

This section is devoted to the investigation of the effect of pristine and functionalized MWCNTs on mature biofilms (Figure 2).

It was shown that the biofilms of *R. ruber* gt1, *A. faecalis* 2, and *A. guillouiae* 11 h were much more destroyed under oMWCNTs than those in the control, and the biofilms of *A. guillouiae* 11 h were also destroyed under hMWCNTs. At the same time, the biofilms of *R. erythropolis* 4-1 in the medium with pMWCNTs, oMWCNTs, and cMWCNTs, as well as *P. fluorescens* C2 in the medium with cMWCNTs and hMWCNTs, and *B. dolosa* PNOS in the medium with hMWCNTs were shown to be highly preserved. Compared to pMWCNTs, the biofilms of *R. ruber* gt1, *A. faecalis* 2, and *A. guillouiae* 11 h were much more destroyed under oMWCNTs. hMWCNTs had different effects on the mature biofilms of different bacterial species. The biofilms of *R. erythropolis* 11-2, *P. fluorescens* C2, and *B. dolosa* PNOS were less susceptible to destruction than upon exposure to pMWCNTs, and the biofilms of *A. faecalis* 2 and *A. guillouiae* 11 h were more strongly destroyed than under pMWCNTs. Complete eradication of the biofilms under the effect of MWCNTs was not observed, and the minimum of the biofilm biomass was observed when *A. faecalis* 2 was exposed to oMWCNTs. Gram-positive actinobacterial biofilms were less susceptible to destruction. Bacterial biofilms were more strongly destroyed when exposed to oMWCNTs.

### 3.3. Effect of pMWCNTs and Functionalized MWCNTs on the Respiratory Activity of Biofilms

The respiratory activity of the biofilms was assessed by interaction with the XTT reagent. The change in the respiratory activity of the biofilms during growth in the presence of pristine and functionalized MWCNTs was studied. The index I_deg_ = OD_exp_/OD_cont_ was calculated. It was shown that the I_deg_ exceeded 1 after the formation of biofilms when exposed to all MWCNTs. This fact indicates that a greater number of viable cells in the biofilms formed in the presence of MWCNTs (Table 1). The exceptions were the biofilm of *R. ruber* gt1 formed under oMWCNTs and that of A. pulmonis PNOS formed under hMWCNTs. The I_deg_ values were slightly lower than the control.

The respiratory activity of biofilms grown on LB medium, which were then exposed to MWCNTs, was determined (Table 1). It was shown that the dehydrogenase activity decreased in *R. erythropolis* 4-1, *A. guillouiae* 11 h, and *A. faecalis* 2 exposed to pMWCNTs and hMWCNTs, as well as in *A. guillouiae* 11 h exposed to cMWCNTs. It was determined that pMWCNTs, hMWCNTs, and cMWCNTs resulted in a significant decrease in respiratory activity when measurements were made in the presence of CNTs for 7 h. *R. erythropolis* IL BIO was the most resistant bacteria, and cMWCNTs had the greatest inhibitory effect.

### 3.4. The Effects of pMWCNTs and Functionalized MWCNTs on the Metabolic Activity of Biofilms, Estimated by the Reduction of Resazurin to Rezarufan

The metabolic activity of the biofilms was assessed by the reduction of resazurin (PrestoBlue^TM^ Cell Viability Reagent) to fluorescent rezarufan. It was shown that oMWCNTs had the least damaging effect on the cells in the biofilms. Not only functionalized MWCNTs but also pMWCNTs reduced the metabolic activity of strains. The metabolic activity of Ac. guillouiae 11 h and Al. faecalis 2 decreased slightly after exposure to pMWCNTs (Table 2). The metabolic activity of bacteria was greatly reduced by hMWCNTs and cMWCNTs.

## 4. Discussion

During the formation of biofilms in the presence of pristine and functionalized MWCNTs, we observed both an increase and a slight decrease in biofilm formation. The response to the presence of MWCNTs in the medium was strain-dependent. MWCNTs in the medium had a lesser effect on Gram-positive bacteria than on Gram-negative bacteria. In the presence of MWCNTs, Gram-negative bacteria formed a biofilm biomass that was 4 times greater than that of the control. The enhancement of biofilm formation may be an adaptive response of the bacteria to unfavorable external factors [45]. Additionally, the presence of CNTs in the culture medium can promote bacterial aggregation and enhance adhesion to the surface. The motile cells of Gram-negative bacteria aggregated with MWCNTs and more easily settled on the surface and adhered to it or to a growing biofilm. No complete inhibition of biofilm formation was observed in the presence of pristine and functionalized MWCNTs, even when exposed to hMWCNTs and cMWCNTs. So, it can be concluded that these functionalized MWCNTs do not have any significant antibacterial effect.

Complete eradication of the biofilms under the influence of pristine and functionalized MWCNTs was not observed. The biofilms of Gram-negative bacteria were more susceptible to destruction by nanomaterials than the biofilms of Gram-positive bacteria. Due to its structure, the EPS matrix can trap and accumulate nanoparticles [15]. Several stages can be distinguished in the interaction of nanoparticles with bacterial biofilms, including the transfer of nanoparticles to the biofilm, attachment to the surface of the biofilm, and migration inside the biofilm [35]. The adsorption of CNTs on the biofilm surface occurs during the deposition of nanotube aggregates under the influence of gravity. During migration in the polymer matrix, nanotubes lead to an increase in the process of biofilm erosion and detachment of its parts. The motile cells of Gram-negative bacteria migrated more actively from the biofilm, increasing its destruction. In *P. fluorescens* C2, on the contrary, significantly greater preservation of the biofilm was noted after exposure to hMWCNTs and cMWCNTs. In this case, the MWCNTs acted as a “cementing” material that prevented the destruction of the biofilm.

The metabolic activity of the biofilm cells was assessed by two methods. The first method was based on the cleavage of the yellow tetrazolium salt XTT to form an orange formazan dye by metabolic active cells. The tetrazolium salts are reduced by dehydrogenase enzymes present in the electron transport systems of cells. An increase in the number of living cells resulted in an increase in the overall activity of dehydrogenases in the sample. This increase directly correlated to the amount of orange formazan formed. An increase in OD 480 during the reduction of XTT by cells of the biofilms formed in the presence of MWCNTs may indicate a greater number of viable cells in the composition of the biofilms. MWCNTs in the culture medium did not reduce cell viability and did not have an antimicrobial or antibiofilm effect. The effect of MWCNTs on mature biofilms, in most cases, also increased the dehydrogenase activity, which could be a consequence of the cell’s response to a stress factor. A decrease in the amount of formed formazan was observed after exposure to hMWCNTs on the biofilms of Gram-negative bacteria, which may indicate a decrease in the level of metabolism. A significant decrease in XTT reduction in the presence of CNTs can be explained by the direct effect of CNTs on this process; however, the I_deg_ was close to 1 in a number of strains under exposure to pMWCNTs, oMWCNTs, and hMWCNTs. Although a direct influence of CNTs on electron transfer cannot be ruled out, if the electron transfer process was completely suppressed, there would be no exceptions among the strains in which the decrease in dehydrogenase activity was insignificant. The duration of contact between the CNTs and cells in this variant of the experiment was longer, which could have enhanced the damaging effect of the CNTs.

The second method of assessing the metabolic activity was to determine the level of the reduction of resazurin to fluorescent rezarufan. Resazurin is blue and non-fluorescent in its oxidized state but is reduced to a pink fluorescent derivative, resorufin, by any reductase enzymes [46]. It was shown that the metabolic activity decreased to the greatest extent under pMWCNTs, hMWCNTs, and cMWCNTs, while oMWCNTs had a less damaging effect. Carboxyl and hydroxyl groups in the compositions of nanotubes increase the affinity of these nanomaterials for the surface structures of the cell, which can lead to an effect on the membrane, uncoupling the membrane potential, and a negative effect on the cell energy. Different studies have shown antimicrobial properties of MWCNTs, but the antibacterial activity of MWCNTs is significantly lower than that of SWCNTs [47]. Hydroxyl and carboxyl functionalized MWCNTs in suspensions did not show antimicrobial activity to *Salmonella typhimurium*, *Bacillus subtilis*, and *Staphylococcus aureus* cells at their concentrations up to 500 µg/mL [48]. However, hydroxyl and carboxyl functionalized MWCNTs are used to create composite materials with antifouling effects. Gholami et al. reported on the improved antifouling feature of polyvinylidene fluoride membrane after it was modified by hydroxyl and carboxyl functionalized MWCNTs. BSA was used as a model to study fouling tests [49]. Alizadeh et al. showed that *Escherichia coli* and *S. aureus* grew on composite membranes coated with cMWCNTs, but less than 4% of the living cells remained, and the biofilm was washed off the membrane due to water shear forces [32].

The results of the two methods of assessing metabolic activity did not correlate with each other. When exposed to CNTs for 1 h, followed by washing and adding XTT, no significant decrease in the dehydrogenase activity was shown. The determination of the optical density of the samples containing XTT occurred within 7 h. At this time, the cells could restore their physiological functions if the effect of the factor was not lethal. With longer exposure, the XTT assay showed a decrease in the dehydrogenase activity, as did the resazurin assay. This was possibly due to the greater accuracy of fluorescent analysis compared to colorimetric analysis and a faster response. An increase in the dehydrogenase activity could be associated with the cell’s response to stress caused by the presence of nanomaterials or with the effect of carbon nanotubes on the membrane. A decrease in the amount of formazan could be caused by a negative effect on the respiratory chain and cell metabolism as a whole. Alonso et al. compared the XTT and resazurin assays for quantification of the metabolic activity of *S. aureus* biofilm and reported no correlation between the XTT and resazurin assays in this determination [50]. Thus, these methods for studying biofilms do not exclude but complement each other. In this case, the analysis of resazurin is faster but requires the detection of fluorescence.

The effect of MWCNTs on the respiratory activity of biofilms depends on the modification of MWCNTs and on the concentration of MWCNTs in the medium. It is known that dehydrogenase activity slightly increases with the addition of 5 mg/g of MWCNTs and decreases in the presence of 50 mg/g of MWCNTs [13]. In our experiments, 200 mg/L MWCNTs affected the biofilms, and the maximum decrease in respiratory activity (up to 53–55%) was found in the mature biofilms of *A. faecalis* 2 and *A. guillouiae* 11 h under the influence of pMWCNTs and hMWCNTs, respectively.

The effect of CNTs on activated sludge requires further study. It has previously been shown that the microbial viability and metabolic activity of active sludge were significantly compromised in the presence of graphene oxide (the functionalized form of graphene with epoxy, hydroxyl, and carboxyl groups) with a concentration higher than 100 mg/L. The formation of the biofilm was significantly compromised in the presence of the multilayer nano-graphene oxide at 300 mg/L, while the acute exposure at this dosage caused the dissociation of the mature biofilm [51]. Our study shows that the biofilms of Gram-negative bacteria of the active sludge were more susceptible to 200 mg/L CNTs than those of the Gram-positive soil, and hMWCNTs and cMWCNTs had the greatest effect among the CNTs. Biofilm formation is an adaptive response of bacteria. It can be concluded that cMWCNTs and hMWCNTs promote biofilm formation but will inactivate some biofilm cells due to a damaging effect. It is known that the presence of MWCNTs with OH and COOH functional groups on modified membranes enhances bacterial absorption [32]. We investigated the effect of functionalized MWCNT dispersion on biofilm formation and showed that cMWCNTs and hMWCNTs enhanced biofilm formation in Gram-negative bacteria but, later, negatively affected their metabolism.

## 5. Conclusions

Thus, the effect of functionalized MWCNTs on biofilms depended on whether the bacteria were Gram-negative or Gram-positive. The biofilms of Gram-negative bacteria were more prone to destruction when exposed to MWCNTs, but their biomass was greater when they were grown in the presence of functionalized MWCNTs. Of all the modified carbon nanotubes, hMWCNTs and cMWCNTs had the greatest effect on the formation of Gram-negative bacteria biofilms, and oMWCNTs had the greatest effect on degradation. None of the studied carbon nanotubes resulted in the complete inhibition of biofilm formation or the complete eradication of mature biofilms; hMWCNTs and cMWCNTs had the greatest effect on the metabolic activity, while oMWCNTs had the smallest effect. The effect of MWCNTs was strain-dependent. Carbon nanotubes at a high concentration and modified with carboxyl and hydroxyl groups can lead to a decrease in the effectiveness of the ability of useful biofilms to biodegrade pollutants both in activated sludge communities and in soils. At the same time, the modification of carbon nanotubes by these groups can be used to create composite materials with an antifouling effect.

## Figures and Tables

**Figure 1 microorganisms-10-01627-f001:**
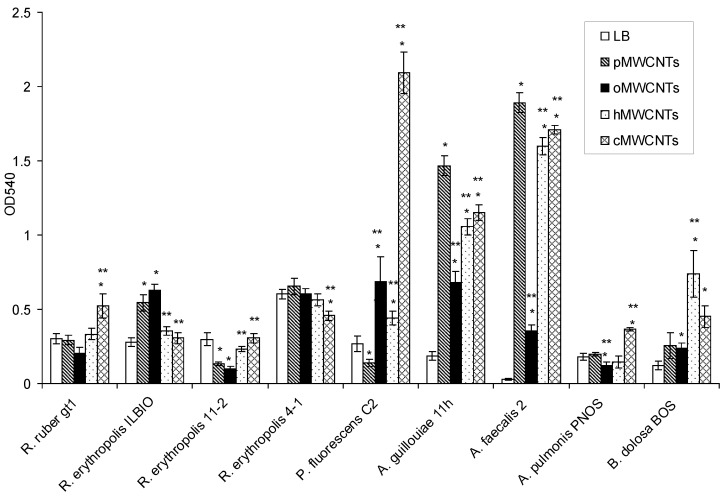
Biofilm formation on the medium: LB, LB with pMWCNTs, oMWCNTs, hMWCNTs, cMWCNTs. * Difference between biofilm formation on LB and LB with CNTs, *p* < 0.05; ** difference between biofilm formation on LB with pMWCNTs and LB with functionalized MWCNTs, *p* < 0.05.

**Figure 2 microorganisms-10-01627-f002:**
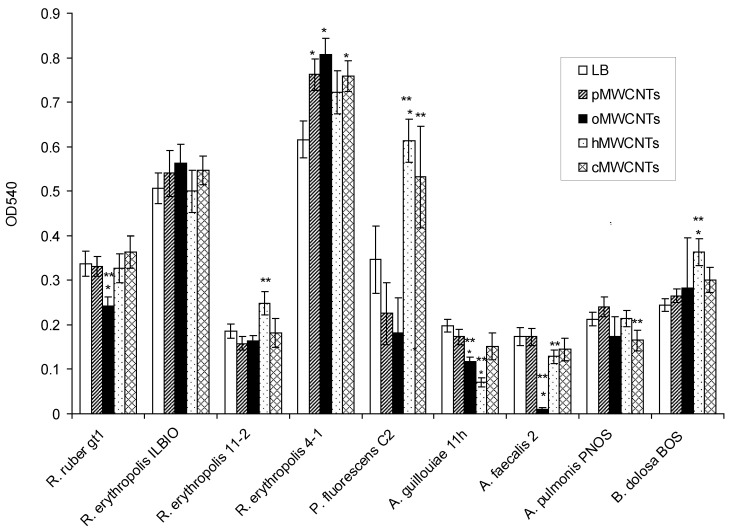
Eradication of biofilms under the influence of 0.9% NaCl solution, 0.9% NaCl solution with pMWCNTs, oMWCNTs, hMWCNTs, and cMWCNTs. * Difference between biofilm eradication under 0.9% NaCl (control) and biofilm eradication under MWCNTs, *p* < 0.05; ** difference between biofilm eradication under pMWCNTs and functionalized MWCNTs, *p* < 0.05.

**Table 1 microorganisms-10-01627-t001:** Effects of pMWCNTs and functionalized MWCNTs on the respiratory activity of biofilms, I_deg_.

Strains	pMWCNTs			oMWCNTs			hMWCNTs			cMWCNTs		
	a	b	c	a	b	c	a	b	c	a	b	c
*R. ruber* gt1	1.70 *	1.50 *	0.09	0.78 *	1.51 *	0.83	2.19 *	1.30	0.21	1.83 *	1.97 *	0.12
*R. erythropolis* IL BIO	1.50 *	1.47	0.91	2.30 *	1.08	0.66 *	1.57 *	1.09	0.91	2.48 *	1.55 *	0.13 *
*R. erythropolis* 11-2	1.57 *	1.21	0	1.34 *	1.78 *	0.70	1.43	1.01	0	1.14	1.30	0
*R. erythropolis* 4-1	2.59 *	0.67	0.22 *	2.78 *	1.09	0.40	1.66 *	0.63	0.39	1.99 *	1.09	0
*Ac. guillouiae* 11h	2.05 *	0.60 *	0	1.30 *	0.89	0.61 *	2.27 *	0.47 *	0	1.98 *	0.79	0
*Al. faecalis* 2	2.08	0.45	0	1.20	0.96	0.29	1.20	0.55	0	1.13	1.35	0
*P. fluorescens* C2	1.19 *	1.27	0.27 *	0.94	1.25 *	0.37 *	1.25 *	0.86	0	1.15 *	2.05 *	0
*A. pulmonis* PNOS	1.24 *	0.58 *	0	1.29 *	1.49	0.83	0.87 *	0.62 *	0.13 *	1.12 *	1.20	0
*B. dolosa* BOS	1.04	1.04	0.38 *	1.42 *	1.85 *	0.81	0.97	3.13 *	0.88	1.03	2.15 *	0

Biofilms grown in the presence of CNTs (a), effect of CNTs on mature biofilms, measurements for 7 h after 1 h exposure to CNTs (b), effect of CNTs on mature biofilms, measurements for 7 h in the presence of CNTs (c); I_deg_ > 1—increase in respiratory activity; I_deg_ < 1—decrease in respiratory activity; I_deg_ = 1—no effect (* *p* < 0.05).

**Table 2 microorganisms-10-01627-t002:** Metabolic activity of biofilms after exposure to pristine and functionalized MWCNTs, %.

Strains	pMWCNTs	oMWCNTs	hMWCNTs	cMWCNTs
*R. ruber* gt1	37.3 *	75.6	58.3	29.9 *
*R. erythropolis* IL BIO	30.3 *	84.4	29.0 *	58.2 *
*R. erythropolis* 11-2	8.9 *	77.2	25.8 *	60.1
*R. erythropolis* 4-1	45.6 *	116.5	26.8 *	49.8 *
*Ac. guillouiae* 11 h	87.3	142.1	42.5	56.6 *
*Al. faecalis* 2	90.8	123.5	61.1	72.5
*P. fluorescens* C2	15.7 *	127.7	19.9 *	19.7 *
*A. pulmonis* PNOS	23.8 *	64.6 *	33.8 *	33.8 *
*B. dolosa* BOS	4.8 *	49.3 *	44.7 *	14.5 *

The control without exposure to MWCNTs was taken as 100% (* *p* < 0.05).

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
