# Peer review of "Functionalization of Multi-Walled Carbon Nanotubes Changes Their Antibiofilm and Probiofilm Effects on Environmental Bacteria"

_microorganisms, 2022, doi:10.3390/microorganisms10081627_

Round 1
Reviewer 1 Report
The manuscript by Maksimova et al. deals with the important issue of carbon nanomaterials influence on the environmental bacteria. Functionalization of MWCNTs can be a good method to change their physicochemical properties and biological activity. The manuscript, although interesting, has some serious shortcomings and seems to be very preliminary. In this form it cannot be published in the reputable "Microorganisms" journal.
Major remarks:
1/ the Authors present 4 experiments on: biofilm formation and eradication, as well as respiratory and metabolic activity of bacteria strains exposed to different MWCNTs. Why do the same materials (e.g. cMWCNTs) stimulate formation and eradication at the same time? How to explain that phenomenon?
2/ the same problem concerns respiratory and metabolic activity of bacteria. Should not they be the same? How to explain that cMWCNTs increase the respiratory activity of R. ruber but decrease their metabolism?
3/ the Authors cite Alonso et al. for explanation that XTT and rezasurin tests are not comparable but both assays depend on the mitochondrial dehydrogenases activity. Maybe additional tests would help to explain these inconsistencies.
3/ considering the abovementioned remarks, there is no clear conclusion drawn from these experiments. Are the modifications of MWCNTs beneficial or not? What kind of modification would help to grow the biofilms and what would impair the biofilm formation? More experimental data are required to increase the quality of this study
Minor remarks:
1/ in Figures 1 and 2 the Authors should include the legend instead of numbers above the bars.
Author Response
Dear reviewer, thank you for considering our manuscript and for your valuable comments. Additional information is included in the text of the manuscript and marked in green.
1/ the Authors present 4 experiments on: biofilm formation and eradication, as well as respiratory and metabolic activity of bacteria strains exposed to different MWCNTs. Why do the same materials (e.g. cMWCNTs) stimulate formation and eradication at the same time? How to explain that phenomenon?
Stimulation of biofilm formation can be the result of several processes: adaptation to unfavorable factors and enhancement of the process of aggregation and adhesion of cells to the surface, due to the presence of a material such as CNTs in the medium. The destruction of a biofilm can be a mechanical process stimulated by the presence of CNT aggregates in the environment, as well as a natural process that occurs when nutrients are depleted in the environment. The processes of biofilm formation and destruction of biofilms can be caused by different mechanisms. In addition, the destruction of biofilms was observed in experiments with mature biofilms, and biofilm formation is a more complex, dynamic process, which is influenced by various factors at different stages of biofilm maturation. This may be an increase or decrease in adhesion at the stage of initiation of biofilm formation, an effect on cell viability, the possibility of biofilm growth on dead cells, and biofilm destruction during its maturation. Therefore, in our opinion, if one of the types of MWCNTs increases biofilm formation, it does not necessarily contribute to less destruction of the mature biofilm.
2/ the same problem concerns respiratory and metabolic activity of bacteria. Should not they be the same? How to explain that cMWCNTs increase the respiratory activity of R. ruber but decrease their metabolism?
Respiratory and metabolic activity should be comparable, but the methods by which they are assessed are indirect. The method for assessing metabolic activity using resazurin is faster, since it is based on fluorescence detection, the reagent reacts almost immediately. An increase in the optical density of samples containing XTT occurs within 7 hours. During this time, cells can restore their functions if the effect of the factor was not lethal. Therefore, we believe that the method based on the reduction of resazurin is more accurate. We justify this in the Discussion, additions are highlighted in green.
3/ the Authors cite Alonso et al. for explanation that XTT and rezasurin tests are not comparable but both assays depend on the mitochondrial dehydrogenases activity. Maybe additional tests would help to explain these inconsistencies.
Indeed, the reducing XTT and resazurin occurs by dehydrogenases. We have done an additional test to assess the effect of CNTs on the recovery of ХTT, in which the optical density was measured in the presence of CNTs for 7 hours. This showed a significant decrease in respiratory activity. With longer exposure, the XTT assay shows a decrease in dehydrogenase activity, as does the resazurin assay. Although a direct influence of CNTs on electron transfer cannot be ruled out, if the electron transfer process was completely suppressed, there would be no exceptions among strains in which the decrease in dehydrogenase activity is insignificant. The duration of contact between CNTs and cells in this variant of the experiment is longer, which can enhance the damaging effect of CNTs. But in the test with resazurin, the reaction is faster. This is possibly due to the greater accuracy of fluorescent analysis compared to colorimetric analysis, and faster response.
4/ considering the abovementioned remarks, there is no clear conclusion drawn from these experiments. Are the modifications of MWCNTs beneficial or not? What kind of modification would help to grow the biofilms and what would impair the biofilm formation? More experimental data are required to increase the quality of this study.
We consider the effect of CNTs on biofilms of environmental bacteria, that is, we want to answer the question of whether the emission of CNTs into the environment in large volumes will adversely affect the biofilm formation of beneficial bacteria, including those in activated sludge. From the results obtained, it can be concluded that, on the one hand, cMWNTs increase the biofilm formation of Gram-negative bacteria, and, on the other hand, they reduce metabolism. This is not a contradiction, since the transfer from the planktonic mode to the attached one is an adaptive response, and in a biofilm, some cells can either die or be in a persistent state, which contributes to the survival of the rest of the population.
Additional experimental data are included in the Results, Table 1 and Discussion.
Minor remarks:
1/ in Figures 1 and 2 the Authors should include the legend instead of numbers above the bars.
Correction was made.
Reviewer 2 Report
The manuscript entitled "Functionalization of Multi-Walled Carbon Nanotubes Changes Their Antibiofilm and Probiofilm Effects on Environmental Bacteria" deals with the effect that multi-walled carbon nanotubes have on bacterial biofilms.
To this purpose the authors investigated the effect of functionalized MWCNTs on biofilm formation and viability of proteobacteria and of actinobacteria in biofilms. They found that the aforementioned influence depends on whether the bacteria are Gram-negative or Gram-positive in respect to the degree of destruction and their biomass. Among the differently functionalized nanotubes carboxylated MWCNTs and hydrophilic MWCNTs show different characteristics, probiofilm and antibiofilm effects than oleophilic MWCNTs. Authors present the data and analyse them appropriately coming to some interesting results which can be a start for further examination.
The authors could try to go deeper to the underlying mechanisms.
Nevertheless, the manuscript poses a question properly defined while results are consistent with the evidence.
In conclusion, I evaluate that the manuscript is suitable for publication.
Author Response
Dear Reviewer, thank you for the positive review of our manuscript. In the future, we will study the mechanisms of these processes.
Round 2
Reviewer 1 Report
The Authors have answered my comments and addressed all the remarks. However, no additional experiments were performed, thus I still consider this manuscript quite preliminary. I leave the final decision to the Field Editor.
Author Response
Dear Reviewer,
Thank you for reviewing our manuscript. As an additional experiment, XTT reduction in the presence of carbon nanotubes was added. This experiment showed a significant decrease in biofilm cell metabolism, as did the resazurin analysis.